# Identifying Key Assessment Factors for a Company's Innovation Capability Based on Intellectual Capital: An Application of the Fuzzy Delphi Method

Benny Lianto 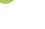

Industrial Engineering Department, Universitas Surabaya, Surabaya 60284, Indonesia; b_lianto@staff.ubaya.ac.id

**Abstract:** Innovation has become one of the most important sources of a company's sustainable competitiveness. Therefore, every company strives continuously to improve its innovation capability. A company's innovation capability is largely determined by various factors originating from its tangible and intangible resources. A lot of research related to increasing innovation capability with assessment factors originating from tangible resources has been carried out, whereas the use of assessment factors originating from intangible capital, such as intellectual capital, is still relatively limited. This study aims to identify and screen the key assessment factors for innovation capability, based on the intellectual capital of Indonesia's manufacturing sectors. This study used a systematic literature review and focus group discussions to establish 18 initial assessment factors, after which 14 final factors were screened out by industry practitioners and academic experts using the Fuzzy Delphi Method. Four factors had the highest de-fuzzy value (0.89), namely adaptation, innovation behavior, organization culture, climate, and forward linkages. The results also showed that three factors of the aspects of human capital (adaptation, innovation behavior, and high motivation and commitment) are considered important determinants for the assessment of innovation capabilities, based on intellectual capital.

**Keywords:** innovation capability; intellectual capital; sustainable competitiveness; manufacturing

## 1. Introduction

Nowadays, with the very competitive business environment, innovation has become one of the most important sources of competitive advantage for all industrial sectors [1–4]. Lee and Trimi [3] stated that sustainable innovation is imperative for organizational survival and success in the turbulent market environment of the digital age, especially more so in the current COVID-19 pandemic crisis. Many studies show that sustainable innovation and corporate innovation capability are essential for a company to achieve sustainable competitiveness [5–9]. The capability for innovation shows the potential of companies to create innovations, and this relates to them having the resources and abilities to create innovative results. These factors are also called the assessment factors for innovation capability [10]. To maintain its capability to innovate, every company needs to know its vital assessment factors for innovating, so that each company can immediately determine its priorities and the strategic steps for developing these key factors effectively and sustainably [11]. A company's innovation capability is largely determined by various factors originating from its tangible and intangible resources.

Various models that assess innovation capabilities have been developed and applied to multiple companies [12–16]. However, the models are more likely to use assessment factors for innovation capability from the aspect of tangible or financial capital, such as the number of employees, number of machines, technology, and total R&D costs. The utilization of intangible capital, such as intellectual capital and knowledge, is still relatively limited. In contrast, in today's era of knowledge-based economies and rapid technological changes,

Inkinen [17] highlighted that the primary source of value creation in a company has shifted from tangible production factors to intangible resources. Alvino et al. [18] also mentioned that a company's value is currently largely determined by its intangible intellectual capital. Intellectual capital refers to the knowledge stock embedded in human, organizational, and relational resources, as well as the activities of the organization [19]. Mirela Nichita, Ref. [20] found that "intellectual, not physical capital, is the most important asset of a company." These findings are in line with Ornek and Ayas [21] who stated that "while the financial capital became influential within the businesses, the intellectual capital concept began to come into prominence." Previous studies also presented a strong link between intellectual capital and the capability to innovate [22–31]. With that shift, a company needs to redefine and change its choice of crucial factors for assessing its innovation capability. Cassol et al. [32] also mentioned that companies need to utilize intellectual capital as one of the essential factors capable of mobilizing a firm's innovation potential.

This study aims to identify and screen the key assessment factors for innovation capabilities based on intellectual capital in Indonesia's manufacturing sector, using the Fuzzy Delphi Method (FDM). The research questions developed in this study are: "What are the assessment factors for innovation capabilities based on intellectual capital that can be used in manufacturing industry in Indonesia and how are these key assessment factors selected?" Several studies on innovation capability related to assessment factors originating from intellectual capital have been conducted [33–35]. However, generally, there are only three aspects of intellectual capital that are used, namely: human capital, structural capital, and social capital. This study uses more comprehensive and detailed aspects of intellectual capital, namely: human capital (HMC), technoware structural capital (TSC), infoware structural capital (ISC), organizational structural capital (OSC), and social capital, which have been subdivided into inter-organizational social capital (ESC) and intra-organizational social capital (ASC). Specifically for structural capital, this study has adopted the Technology Atlas Project's approach [36] by identifying three separate structural components, namely: (1) technoware (object embodied technology), (2) infoware (document embodied technology), and (3) orgaware (institutional embodied technology).

The manufacturing sector was chosen because its activities provide a broad chain effect on the national economy [37]. The results of this study aim to provide evidence about closing the limitation gap for assessing innovation capability based on intellectual capital and give critical inputs for company managers to understand the key assessment factors for innovation capability derived from intangible resources, so they can be monitored effectively and sustainably to achieve the company's sustainable competitiveness.

## 2. Theoretical Background

Innovation capabilities are associated with how firms systematically generate and modify processes to integrate and reconfigure their internal and external resources, which leads to improved effectiveness in times of rapid global technological change [38]. Innovation capability is also considered to be the capacity of a firm to generate new ideas, seize opportunities from various sources, and use all these potential aids to drive innovation [39]. From a similar perspective, Ngo et al. [40] stated that innovation capability includes all the characteristics and assets of the company that can be used to facilitate and support the company's innovation strategy. All resources, potentials, and assets owned by the company can be referred to as the assessment factors for innovation [10]. Assessment factors for innovation of a company can be sourced from its tangible and intangible resources. A lot of research related to increasing innovation capability with assessment factors originating from tangible resources has been carried out, whereas the use of assessment factors originating from intangible capital, such as intellectual capital, is still relatively limited. In fact, previous studies presented a strong link between intellectual capital and the capability to innovate [30,31]. Li et al. [41] stated that an organization with a greater amount of intellectual capital has a stronger innovation capability. Intellectual capital is closely related

to the level of production and the flow of knowledge within an organization, so it affects the capability to innovate [42].

Ali et al. [43] define intellectual capital as the aggregate sum of knowledge that an organization can utilize to improve its performance and competitive advantage. Gomez-Valenzuela [44] conceptualizes intellectual capital as a body of knowledge or know-how that can be utilized by a company to increase its value. Many other scholars define intellectual capital as a set of intangible assets, capabilities for creating assets, and social relationships that create value for an organization [44–46]. Although there is no precise definition of intellectual capital, most scholarly articles describe that intellectual capital encompasses human capital, structural capital, and social capital [47]. Human capital refers to the members of an organization and their level of education, learning capabilities, knowledge, skill, creativity, loyalty, leadership, motivation, attitude, and values [41,48]. On the other hand, Martinindis et al. [30] opined that human capital refers to the value of human capacity. Previous studies stated that human capital is the primary and vital resource of an organization, as without it, knowledge cannot be generated and developed [31,49] and organizations cannot achieve anything (including innovation) [41,50]. Structural capital refers to the stock of assets and an organization's mechanisms that support the members of the organization in activities to transform ideas and innovations into real properties [51]. Li et al. [41] argue that structural capital encompasses "all non-human storehouses" (e.g., databases, organizational structures, work manuals, strategies, and procedures). Additionally, structural capital involves information systems, operational flows, culture, policies, patents, copyrights, etc. [48]. Prior studies have recognized that structural capital plays a vital role in transforming external knowledge into various forms of policies and strategies within a company, which are very useful for improving the innovation process [46]. Relational capital refers to a company's ability to accumulate value from collaborative relationships with various external parties [52]. Al-Khatib [31] stated that relational capital describes the human ability to interact with the external environment and the ability to learn from the experiences of others so that new ideas can be generated that will improve the company's innovation capabilities and performance. On the other hand, Li et al. [41] opined that relational capital is a very close interpersonal relationship based on trust, commitment, and respect for all internal and external stakeholders. Based on this close relationship, a company's human resources get a lot of information, knowledge, and opportunities which can further improve the company's ability to innovate. According to Ahmed et al. [53], good relationships and good cooperation networks with internal and external stakeholders are more important than resources.

Several previous studies on the manufacturing industry have shown that all three elements of intellectual capital have a significantly positive effect on innovation capability. There follows a description of the assessment factors for innovation capabilities based on intellectual capital that is relevant to the manufacturing industry:

### 2.1. Competence

The level of competence of human resources as the company's human capital in this study is measured by education level, number of certifications, and amount of training [54]. HR competence is widely observed as one of the determining aspects for a manufacturing company to develop its innovation capability [55,56]. Human resources that have skills and a high level of education and regularly attend training have contributed to supporting companies in generating intellectual property rights and are motivated to provide suggestions and input, often for the product innovation development process [54]. On the other hand, Mir Dost et al. [57] stated that employees who are highly skills and advanced knowledge will always be a source of new ideas for the development of innovation.

### 2.2. Innovation Behavior

Innovation behavior in this study, which describes employee creativity to generate new ideas, is measured by the number of new product ideas and processes sourced from



employees. Several studies in the manufacturing industry show that highly innovative behavior by employees will create and encourage the formation of a healthy climate for giving birth to a variety of new product development ideas [24,58,59]. Al-Zu'bi [59] stated that high employee innovation behavior shows the ability of employees to seek and develop new ideas, and then communicate and implement those new ideas into various product innovations. This also shows that employees who have good innovation behavior tend to try to realize their ideas by attempting to overcome various problems in the innovation development process.

### 2.3. High Motivation and Commitment

The high commitment and motivation of employees in this study are measured by the level of participation in innovation activities. Research on efforts to increase innovation capability in the manufacturing industry has found that high employee commitment and motivation will cause them to have high participation and involvement in all innovation activities in the company [60,61]. Ben Moussa and El Arbi [56] stated that it is impossible for employees' creativity and innovation capabilities to develop if they do not have high commitment and motivation toward their work.

### 2.4. IT Resources

The capabilities of IT resources in this research are measured by the availability and reliability of internet and intranet owned by the company. The factor of IT resources as structural capital has been recognized as a factor that greatly determines a company's innovation capability, especially in the era of digital technology development. Research in several manufacturing industries has shown that IT resources are a factor that determines a company's innovation performance [62,63]. Nieves and Osorio [64] said that IT resources will assist the knowledge integration process within the organization and will further assist the innovation management process. On the other hand, Chen et al. [63] stated that the quality of IT resources really helps manufacturing companies in radically innovating their services.

### 2.5. Technology Flexibility

Technology flexibility describes the capability of production technology owned by the company. In this research, the degree of flexibility in production technology is measured by how well the company's production technology is compatible with the company's innovation strategy. Studies in several manufacturing industries have shown that technology flexibility has a positive effect on a company's innovation capabilities [64–66]. Production technology or manufacturing technology flexibility will enable companies to adopt concepts such as Just in Time (JIT) which, in turn, have a positive influence on the process of developing new product innovations [64].

### 2.6. System and Procedure

In this study, the system and procedure factors for supporting innovation activities are measured by the availability and effective implementation of SOPs and the number of management system certifications. Several studies in the manufacturing industry have shown that system and procedure factors, as part of structural capital infoware, contribute to efforts to increase innovation capability [67,68]. Bernardo [69] stated that the existence of standard systems and procedures that are strengthened by management system certification, for example, the Total Quality Management (TQM) certificate will facilitate the process of adopting innovation because the company already has systems and operation procedures (SOP) for carrying out the innovation management process. Innovation activities managed by a neater and more structured management system will certainly provide maximum results.

### 2.7. Data and Information System

In this research, data and information system factors are measured by the availability and reliability of database systems, data mining systems, and business intelligence. The influence of data availability and information system factors has been shown to have a significant influence on efforts to increase innovation capabilities in the manufacturing industry [62,70,71]. The results of Soto Costa et al. [70], who researched the small and medium-scale manufacturing industries, show that data and information system factors such as structural capital infoware have a positive influence on organizational innovation. The use of e-business systems, for example, greatly assists the distribution process and the process of sharing employee experiences regarding innovation activities with all parts of the organization. In addition, the application of e-business intelligence systems can help companies to continuously improve their processes by monitoring business activity and accessing data analytics timely. Chatterjee et al. [72] stated that a company that has a good database system and has a data-driven culture will find it greatly assists the process of product development and process innovation.

### 2.8. Culture of Innovation

The culture of innovation factor describes the capability of corporate culture in supporting innovation activities measured by the level of freedom with which employees must carry out improvements and innovations. Various studies in several manufacturing industries show that the influence of the culture of innovation factor as organizational structural capital has a very significant influence on efforts to increase innovation capability [73–76]. Chen et al. [73] said that companies that have a strong innovation culture and are compatible with their innovation strategy tend to have a higher speed and quality of innovation. On the other hand, Bendak et al. [77] have demonstrated that corporate culture factors will act as a driving force for innovation in organizations.

### 2.9. Organization Agility

The organizational agility factor describes the capability of organizational cohesiveness in responding to and supporting innovation activities measured by the speed of the decision-making process in the organization. Organizational agility has been widely recognized as an important factor in efforts to increase innovation capability. Studies in several manufacturing industries show that organizational agility encourages increased corporate innovation performance [78–80]. Ravichandran [79] stated that a company's innovation capability has a significant relationship with organizational agility. Companies that have higher innovation capabilities generally have a better ability to leverage their digital platforms to enhance agility. On the other hand, Cai et al. [80] stated that companies with high organizational agility have better product innovation capabilities and performance.

### 2.10. Intellectual Assets

In this study, the intellectual assets factor is measured by the number and application of copyrights, patents, licenses, and trademarks. Intellectual assets have long been recognized as an important factor in efforts to increase innovation performance in the manufacturing industry [81]. Several studies in this sector also show that intellectual assets in the form of the number of intellectual property rights owned by companies (copyrights, patents, licenses, and trademarks) have an impact on innovation performance [82,83]. The results of studies on manufacturing companies in China show that intellectual assets, especially the number of patent applications, have encouraged that country's manufacturing industry to achieve its innovation goals and at the same time support the manufacturing industry to become high-tech in its content manufacturing [83].

### 2.11. Forward Linkage

The forward linkage factor, which is inter-organizational social capital, describes the capability for collaboration with consumers measured by the amount, frequency, and con-

tribution of cooperation. The utilization of consumer knowledge in supporting innovation in the manufacturing industry is commonly practiced [84]. Several previous studies conducted in the manufacturing industry showed that the forward linkage factor, as measured by the company's ability to collaborate with its customers, has a very positive effect on the company's ability to innovate [60,85]. Tavani et al. [86] explained that collaboration by a company with consumers will have an impact on the growth of employee innovation behavior. In addition, collaboration with consumers enables companies to identify customers' unsatisfied needs, thereby enhancing these firms' ability to offer superior products to customers. Companies that cannot cooperate with consumers, especially in being aware of consumer dissatisfaction, will be unable to compete in today's very tight competitive arena.

### 2.12. Backward Linkage

The backward linkage factor, which is part of inter-organizational social capital, describes the capability for collaboration with suppliers measured by the amount, frequency, and contribution of cooperation. Several studies in the manufacturing industry show that the backward linkage factor has a positive influence on increasing the company's innovation performance [60,87,88]. Collaboration based on good mutual trust with suppliers who have high competence has a positive impact on efforts to increase the innovation capability of companies [88]. A study to investigate the effects of suppliers and lead users' collaboration in new product development on innovation behavior in manufacturing companies in EU found that there is a strong positive relationship between the innovation behavior of a company and collaboration with both lead users and suppliers [60]. The results of this study confirm that good quality new ideas in new product development do not only come from internal R&D, but product innovation processes can also be accelerated and strengthened by involving suppliers in innovation activities with the company.

### 2.13. Horizontal Linkage

The horizontal linkage factor describes the capability for collaboration with competitors measured by the amount, frequency, and contribution of cooperation. The horizontal linkage factor has also been demonstrated by several studies in the manufacturing industry to have quite an impact on increasing innovation capabilities [86,89]. Horizontal linkage is also commonly referred to as co-opetition strategy, which is defined as an effort to cooperate while competing with similar companies to gain mutual benefits. Gnyawali and Park [90] stated that co-opetition between strong rivals is a very challenging relationship. The capability to manage this collaboration well will increase results and mutual benefits, especially in terms of advanced technological development and innovation.

### 2.14. Public Linkage

The public linkage factor describes the capability for collaboration with universities and the government as measured by the amount, frequency, and contribution of cooperation. Collaboration with universities, research institutions, and the government has been shown to be one of the determining factors for innovation capability in the manufacturing industry [86,91,92]. Tavani et al. [86] stated that collaboration with external parties in supporting innovation capability improvement in the manufacturing industry shows that only collaboration with research institutions and universities provides significant results in efforts to increase product and process innovation capabilities, provided that the manufacturing industry has a high level of absorptive capacity. In addition, collaboration with research organizations and universities in the context of joint research will increase the increase in knowledge-sharing activities and intensive knowledge transfer within the company and will further trigger an increase in the quality of invention and innovation [93].

### 2.15. Informal Linkage

The informal linkage factor describes the capability for collaboration with professional associations which is measured by the amount, frequency, and contribution of cooperation.

Collaboration with professional associations and informal communities is also one of the determining factors in efforts to increase innovation capabilities in the manufacturing industry [94]. Several studies in the manufacturing industry have shown that informal linkage factors have a significant influence on efforts to increase innovation capabilities [95,96]. The role of professional associations, in some cases, is quite effective as a forum for discussion among similar business actors. This informal friendship model can be used as a way in which to exchange information and ideas for the development of innovation.

### 2.16. Cross-Functional Team

The cross-functional team factor describes the capability for collaboration between functions within the company measured by the number of cross-functional work teams and the number of innovation projects operating between functions. Several studies in the manufacturing industry show that cross-functional team factors have a significant influence on efforts to build a company's innovation capabilities [54,57,65]. Zeng et al. [54] stated that cross-functional integration that occurs within the company will experience a higher speed of new product introduction. Internal collaboration capabilities will improve the quality of interaction and the process of exchanging ideas and ideas between parts of the organization, so, as a whole, this will affect the formation of innovation capabilities [57].

### 2.17. Sharing and Learning

The sharing and learning factor describes the capability for internal communication and interaction collaboration measured by the number of learning and sharing activities. The influence of sharing and learning activities in knowledge management organizations is very significant in efforts to develop new knowledge needed in innovation activities in the manufacturing industry [97,98]. A study in the manufacturing industry shows that social web knowledge sharing that occurs within companies has a significant effect on organizational innovation performance. Soto-Acosta et al. [98] stated that companies that always develop learning and sharing activities internally will create a healthy innovation culture that will greatly affect the formation of long-term innovation capabilities.

## 3. Methodology

This research consisted of two stages:

A.     Determining the assessment factors

The process of determining the initial assessment factors of innovation capability, based on intellectual capital, was carried out in two steps. The first step was to identify the assessment factors by conducting a systematic literature review (SLR). We followed the three-stage procedure of Tranfield et al. [99] that consisted of planning, execution, and reporting. An SLR is an effective method for reviewing collections of research papers or publications. It identifies, selects, and conducts a critical review in order to answer clearly formulated questions. This method must follow clear protocols or plans where criteria are clearly stated before the review is carried out [100]. The process of identifying and selecting initial assessment factors was carried out using the Google Scholar (GS) publication database with search years 2013 to 2022 and search keywords: "Innovation capability" OR "Innovation" AND "Intellectual capital" and papers are written in English (inclusion criterion). There were 78,860 initial papers obtained during this step (searching date 30 August 2022). Three exclusion criteria were then applied to those papers: unchecked citation and title screening (EC1), yielded one international journal that was not reputable and did not contain empirical studies on the manufacturing industry in the abstract section (EC2), and yielded one journal that did not clearly describe the assessment factors regarding innovation capability based on intellectual capital in the contents of the paper (EC3). The article selection process is described in Table 1 as follows:

**Table 1.** An overview of the article selection process.

| No | Inclusion Criteria | Description | Result Searching Date: 30 August 2022 |
|---|---|---|---|
| 1 | Databases | Google Scholar (GS) publication database | |
| 2 | Keywords | "Innovation capability" OR "Innovation" AND "Intellectual capital" | Initial papers: 78,860 |
| 3 | Period | 2013–2022 | |
| 4 | Language | Papers written in English | |
| No | Exclusion Criteria | Description | Result |
| 1 | EC 1 | Unchecked citation and title screening. | 245 |
| 2 | EC 2 | Yielded one international journal that was not reputable and did not contain empirical studies on the manufacturing industry in the abstract section. | 168 |
| 3 | EC3 | Yielded one journal that did not clearly describe the assessment factors regarding innovation capability based on intellectual capital in the contents of the paper. | 43 |

Finally, the paper selection process yielded 43 articles. Next, the assessment factors that would be used in this study were identified and determined. The assessment factors that had been identified in the first step were then evaluated by industry experts through focus group discussions (FGD). The evaluation was carried out to ensure that the initial assessment factors identified from the study of the literature were contextually relevant to the manufacturing sector in Indonesia.

B.    Screening assessment factors

The screening process for the initial assessment factors determined in the first stage was conducted using the FDM. As a collective decision-making method that involves experts, the FDM has been widely employed in diverse cases and sectors [101–105]. In several previous studies, FDM was also used in the process of screening innovation factors and it was stated that FDM is an effective tool in selecting innovation factors [106,107]. There are at least four advantages to the FDM method [108]: (a) overcoming the inevitable uncertainty, (b) reducing the number of surveys, (c) the semantic structure of forecast items can be explained, and (d) the individual attributes of the expert can be described.

Questionnaires were used to gather expert opinions in this study. They were related to the general data and profiles of the respondents and the assessed weight of the importance of pre-determined factors; the extent of the weight used linguistic terms (1–7 scale points) and a fuzzy scale, as seen in Table 2.

**Table 2.** Linguistic variables and fuzzy scale.

| Linguistic Variable | Fuzzy Scale |
|---|---|
| Absolutely Unimportant | (0.0,0.0,0.1) |
| Unimportant | (0.0,0.1,0.3) |
| Slightly Unimportant | (0.1,0.3,0.5) |
| Neutral | (0.3,0.5,0.7) |
| Slightly Important | (0.5,0.7,0.9) |
| Important | (0.7,0.9,1.0) |
| Absolutely Important | (0.9,1.0,1,0) |

A sample of the questionnaire that was used for gathering expert opinions can be seen in Appendix A. The members of the panel of experts involved in this study were industry practitioners and academic experts. The selection of these experts was conducted based on two qualifications: mastery of the research topic (knowledge) and work experience (skill). For the industry practitioners, middle/upper managers with over 10 years of work experience were chosen, as their experience reflects their extensive knowledge and skills.

For the academic experts, a minimum education level of a doctorate was required. Based on the above considerations, 15 experts were chosen (seven academics and eight industry practitioners). Questionnaires containing the pre-determined initial assessment factors for innovation capabilities were distributed to all the experts using a Google Forms survey. Ten experts filled out the form and returned valid responses. Thus, the response rate of this survey was 66.6%, and this was deemed fit as the minimum number of experts required for a panel was achieved [109]. The experts' profiles are shown in Tables 3 and 4.

**Table 3.** List of industrial practitioners.

| Industrial Experts | Position | Sector of Industry |
| --- | --- | --- |
| E1 | CEO | Furniture |
| E2 | GM. Corp. R&D | Food |
| E3 | R&D Manager | Pharmaceuticals |
| E4 | Plant Manager | Bicycle parts |
| E5 | Plant Manager | Aluminum |

**Table 4.** List of academician experts.

| Academic Experts | Area of Expertise |
| --- | --- |
| E6 | Organizational innovation |
| E7 | Industrial engineering |
| E8 | Innovation and technopreneurship |
| E9 | Innovation management |
| E10 | Innovation strategy |

For the process of screening the initial assessment factors, the FDM proposed by Hsu et al. [110] was adopted by this research. The steps for screening the factors using the FDM are depicted in Figure 1.

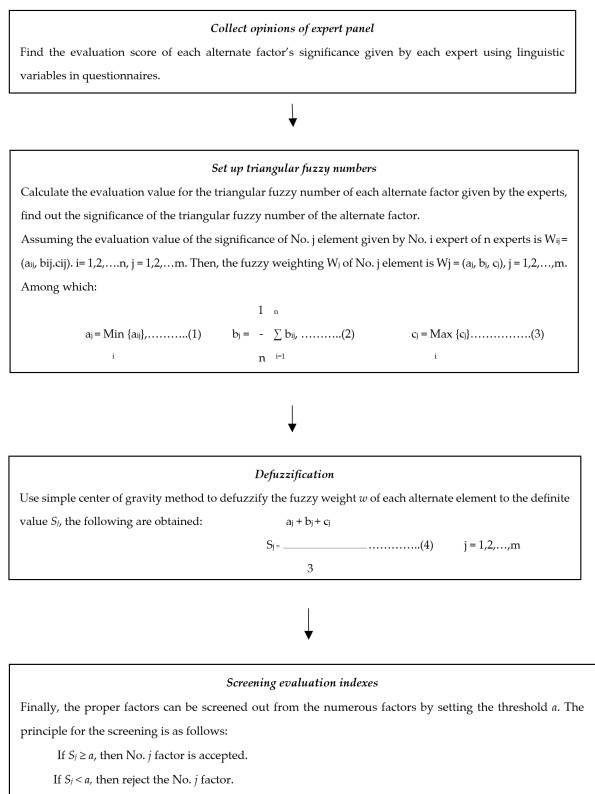

**Figure 1.** Steps of screening the factors using FDM.

## 4. Results

From the process of determining the assessment factors for innovation capabilities based on intellectual capital, 18 initial factors were identified and grouped into six aspects, namely: human capital (HMC), technoware structural capital (TSC), infoware structural capital (ISC), organizational structural capital (OSC) and social capital, which was subdivided into inter-organizational social capital (ESC) and intra-organizational social capital (ASC) [111]. Specifically for structural capital, this study adopted the Technology Atlas Project's approach [36] by dividing the structural components into three components, namely: (1) technoware (object embodied technology), (2) infoware (document embodied technology), and (3) orgaware (institution embodied technology). Eighteen initial factors were chosen, 17 factors were derived from the literature study and one factor was an input from the industry practitioners. They are presented in Table 5.

**Table 5.** Innovation capabilities based on intellectual capital initial assessment factors.

| Intellectual Capital Aspects | Assessment Factors | Description | Sources |
|---|---|---|---|
| Human Capital | Adaptation | The capability of human resources' adaptation in following the development of the external environment and the speed of response to opportunities measured by the number of options manifested into innovations. | Input from industry practitioners |
| | Competence | Company HR capabilities measured by education level, number of certifications, and amount of training. | [54–56] |
| | Innovation behavior | The capability, creativity, and innovation of HR measured by the number of new product ideas and processes sourced from employees. | [24,58,59] |
| | High motivation and commitment | HR capability in contributing to the company is measured by the level of participation in innovation activities. | [56,60,61] |
| Technoware Structural Capital | IT resources | IT resources' capabilities measured by the availability and reliability of internet and intranet owned by the company. | [62,63] |
| | Technology flexibility | The capability of production technology owned by the company and its future development efforts associated with the company's innovation strategy measured with the novelty and flexibility of production technology. | [65–67] |
| Infoware Structural Capital | System and procedure | System capabilities and procedures for supporting innovation activities measured by the availability and effective implementation of SOPs and the number of management system certifications. | [68,112] |
| | Data and information system | Data and information system capabilities measured by the availability and reliability of database systems, data mining systems, and business intelligence. | [63,70,71] |

| Intellectual Capital Aspects | Assessment Factors | Description | Sources |
|---|---|---|---|
| Orgaware Structural Capital | Culture of innovation | The capability of corporate culture in supporting innovation activities measured by the level of freedom employees must carry out improvements and innovations. | [73–76] |
| | Organization agility | The capability of organizational cohesiveness in responding to and supporting innovation activities measured by the speed of the decision-making process in the organization. | [78–80] |
| | Intellectual assets | Capabilities of the company's intellectual assets measured by the number and application of copyrights, patents, licenses, and trademarks. | [82,83] |
| Inter-organizational Social Capital | Forward linkage | The capability of collaboration with consumers measured by the amount, frequency, and contribution of cooperation. | [59,85,86] |
| | Backward linkage | The capability of collaboration with suppliers measured by the amount, frequency, and contribution of cooperation. | [59,87,88] |
| | Horizontal linkage | The capability of collaboration with competitors measured by the amount, frequency, and contribution of cooperation. | [86,89] |
| | Public linkage | The capability of collaboration with universities and the government as measured by the amount, frequency, and contribution of cooperation. | [86,91,92] |
| | Informal linkage | The capability of collaboration with professional associations is measured by the amount, frequency, and contribution of cooperation. | [43,95,96] |
| Intra-organizational Social Capital | Cross-functional team | The capability of collaboration between functions within the company measured by the number of work teams across functions and the number of innovation projects between functions. | [54,57,65] |
| | Sharing and learning | The capability of internal communication and interaction collaboration measured by the number of learning and sharing activities. | [97,98] |

After passing through the four stages of the screening process using the FDM with a threshold value of $\alpha \geq 0.75$ [110], the selected factors are presented in Table 6.

**Table 6.** Key assessment factors after FDM screening.

| Intellectual Capital Aspects | Assessment Factors | Score | | | | Result |
|---|---|---|---|---|---|---|
| | | **Min** | **Max** | **Average** | **De-fuzzy** | |
| Human Capital (HMC) | Adaptation | 0.7 | 1.0 | 0.98 | 0.89 | Accepted |
| | Competence | 0.3 | 1.0 | 0.90 | 0.73 | Rejected |
| | Innovation behavior | 0.7 | 1.0 | 0.96 | 0.89 | Accepted |
| | High motivation and commitment | 0.3 | 1.0 | 0.94 | 0.75 | Accepted |
| Technoware Structural Capital (TSC) | IT resources | 0.5 | 1.0 | 0.95 | 0.82 | Accepted |
| | Technology flexibility | 0.3 | 1.0 | 0.94 | 0.75 | Accepted |
| Infoware Structural Capital (ISC) | System and procedure | 0.5 | 1.0 | 0.94 | 0.75 | Accepted |
| | Data and information system | 0.5 | 1.0 | 0.95 | 0.82 | Accepted |
| Orgaware Structural Capital (OSC) | Organization culture and climate | 0.7 | 1.0 | 0.98 | 0.89 | Accepted |
| | Organization agility | 0.5 | 1.0 | 0.95 | 0.82 | Accepted |
| | Intellectual assets | 0.3 | 1.0 | 0.90 | 0.73 | Rejected |
| Inter-organizational Social Capital (ESC) | Forward linkage | 0.7 | 10 | 0.98 | 0.89 | Accepted |
| | Backward linkage | 0.5 | 1.0 | 0.92 | 0.81 | Accepted |
| | Horizontal linkage | 0.1 | 1.0 | 0.83 | 0.64 | Rejected |
| | Public linkage | 0.5 | 1.0 | 0.94 | 0.81 | Accepted |
| | Informal linkage | 0.3 | 1.0 | 0.90 | 0.74 | Rejected |
| Intra-organizational Social Capital (ASC) | Cross-functional team | 0.5 | 1.0 | 0.91 | 0.80 | Accepted |
| | Media sharing and learning | 0.5 | 1.0 | 0.85 | 0.82 | Accepted |

## 5. Discussion

This study showed the process of identifying and screening the assessment factors for innovation capability, based on intellectual capital, in the context of the manufacturing sectors in Indonesia using the FDM. During the process of identifying the initial assessment factors for innovation capability based on intellectual capital, 18 initial factors were identified, 17 initial factors were derived from the literature study, and one factor derived from the industry experts. All the selected initial factors from the literature review process were confirmed by the industry practitioners through a focus group discussion (FGD) and declared relevant and contextual to the conditions found in East Java's manufacturing industries. Even more interestingly, there was one initial factor input: adaptation, which originated from the industry practitioners' FGD. The FGD results concluded that every company, and especially their human resources, must adapt quickly to any changes that might occur.

After the screening, four factors were rejected: human competence, intellectual assets, horizontal linkages, and informal linkages. Horizontal linkages have the lowest de-fuzzy value of 0.64. Based on the interviews with the experts, specifically, the industry practitioners, building horizontal connectivity with competitors from similar industries was considered unimportant, or even to be avoided. Relationships between similar industries

may occur in the form of involvement in industry associations. However, the results of this study showed that informal linkages described through the participation of companies in industry associations were also considered unimportant.

Another factor considered unimportant as a determining factor for innovation capability was human competence and intellectual assets. Specifically, for the human competence factor, the results were confirmed by several experts who gave a lower assessment of these factors (neutral and a little bit important assessment) and stated that, up to now, the levels of education and skill which were administratively described from the number of diplomas and certificates could not guarantee the ability to work. However, in practice, adaptation, innovation behavior, and high motivation were more important than the level of education and skill. Human resources with high motivation and commitment, who can adapt to changes, and have good innovation behavior, were found to be more valuable compared to administrative measures such as the level of education and the number of skills certifications. Technological progress caused the education level (as measured by diplomas) and skills (measured by the number of certifications and training courses) to be less relevant. In the 4.0 industry era, employees who can quickly learn new things from various sources, with assistance from more advanced technology, are valued more highly by companies.

In comparison, intellectual assets were considered unimportant because, based on follow-up interviews with several experts, many experts stated that intellectual assets such as the number of patents, licenses, copyrights, and trademarks were not very significant for a company. It was essential for a company to have the ability to leverage these assets to foster innovation. In many cases, it has been said that the large number of property rights that cannot be optimally utilized can lead to a cost burden for the company.

Of the 14 accepted assessment factors, four factors had the highest de-fuzzy value: adaptation, innovation behavior, organization culture and climate, and forward linkages. Human adaptation was a factor derived from the industry's input. The focus group discussion resulted in the conclusion that every company, and especially its human resources, must adapt quickly to any changes. This statement agrees with Frizzo-Barker et al. [113], who stated that disruptive technology has changed the rules of the game for everybody. Thus, adaptability is needed by employees and companies to compete in this era of high uncertainty. Hasgall and Ahituv [114] found that adaptability capabilities, as measured by the speed of capturing opportunities and utilizing those opportunities to create excellence, were critical factors facing today's highly dynamic corporate environment.

Another factor with the highest de-fuzzy value was innovation behavior, defined as individual behavior that always tried to introduce new useful ideas and new solutions or procedures in an organization [24]. Liu [115] found that innovation behavior would increase the accumulation process of human capital in supporting innovation activities. The human capital aspects, with three factors from four initial assessment factors, were considered important as determinants of the assessment of innovation capabilities based on intellectual capital, and there were even two factors that had the highest de-fuzzy value. Previous research on innovation capabilities in manufacturing sectors also showed that the factors related to people/human capital were influential in developing innovation capabilities. From an intellectual capital perspective, it can be said that human capital is at the heart of intellectual capital [116]. Banerjee, in the Global Innovation Index report [117], noted that "the human factor is the fundamental driver of innovation."

Overall, based on the analysis in this study, the accumulation of human capital should not be based solely on the level of education and the number of training sessions attended by employees. Adaptability, innovation behavior, and the level of motivation and commitment of employees were found to be more essential for a company to maintain its competitiveness. Liu [115] also found that the accumulation of human capital was primarily determined by the level of motivation and commitment of the employees, and their ability to adapt and capture opportunities, both from the internal environment such as work colleagues,

supervisors, and co-workers, and their relationships with the external environment, such as competitors, suppliers, and consumers.

There were five factors considered as the key assessment factors of innovation capability from the structural capital aspect. Organizational culture and climate were considered the most essential factors with a de-fuzzy value of 0.89. According to numerous studies into innovation capabilities, a conducive culture and organizational climate were both vital factors for a firm's innovation capability [73,118]. A strong organizational culture and climate play a significant role in the formation of a good working environment, enabling employees to express all their ideas and creativity [118]. Fernandes [119] also mentioned that the spirit of an organization's culture was where its employees would always have high motivation to solve every problem faced by the company, and this would be coordinated through the sharing of knowledge and company values. The freedom to do new things without fear and with the full support of top management, and the availability of rewards and incentives for successful innovation, are all examples of a strong innovation culture.

From the perspective of the aspects of social capital, of the five forms of connectivity adopted from Hseih et al. [120], forward, backward, horizontal, public, and informal linkages, only three factors were accepted, while two factors were rejected. Forward linkages, as measured by the number, frequency, and contribution of cooperation with consumers, were the most crucial assessment factors for innovation capability. Costa and Do Vale [121] highlighted that the relationship with the customers, or what is called customer capital, was the most prominent because it was directly related to financial indicators.

In the follow-up interviews with the industry practitioners, most of them stated that a company's collaboration priorities were with the customers, not with the suppliers (backward linkages) or with the educational institutions and the government (public linkages). From the intra-organizational social capital aspect of the two initial assessment factors, cross-functional team, and media sharing and learning, all of them were declared acceptable as key assessment factors for innovation capability. Nowadays, the role of social capital has become increasingly important for improving innovation capabilities. With the advancement of technology, there has been increased connectivity between the individuals within a company, and those outside the company, meaning companies have a great potential to collaborate. By utilizing this connectivity (between team members, between resources), a company can jointly improve its innovation capabilities and generate innovation on an ongoing basis [122].

On the other hand, along with the development of a knowledge-based economic system, the innovation performance of a company can be developed quickly and easily using various sources of knowledge and technology from outside the company [86]. Chayadi Putra et al. [123] found that social capital could help organizations and employees to develop capabilities such as finding new ways to improve operational efficiency.

The results of this study aim to provide evidence of the closing of the limitation gap for assessing innovation capability based on intellectual capital. This study delivered more holistic views of the crucial factors for assessing innovation capability by considering the current landscape of Indonesia's manufacturing industries.

The benefit and practical implication of this study is that it provides insights for manufacturing industry players into how to understand and develop the key assessment factors for innovation capability derived from intangible resources, so they could be monitored effectively and sustainably to achieve a company's sustainable competitiveness. For example, the results of this study recommend that, for now and in the future, with employee performance appraisals it is no longer enough to just look at the level of education alone. Adaptability and innovation behavior are very decisive for the development of human capital, which has an impact on innovation capability. On the other hand, strengthening the innovation culture is a priority and is needed for building structural capital which will further support the development of innovation capabilities. Furthermore, in developing relational capital or social capital, manufacturing industry managers should focus more on strengthening the intra-organizational social capital first. Strengthening

inter-organizational social capital will have an impact on the development of innovation capabilities if the company strengthens its linkages with consumers.

## 6. Limitation and Future Research

This research has several limitations which are as follows: the process of identifying initial assessment factors using the SLR approach only uses one publication database. The Google Scholar database has the advantage of being able to cover a very broad field and it is practical and has no access fees; however, it has lower accuracy than other databases. Another limitation of this study is that the initial assessment factors have been selected from various cases involving similar industries in other countries which are not necessarily in accordance with the conditions found in the manufacturing industry sector in Indonesia. This limitation has been overcome by carrying out the process of verifying the results of the identification of initial assessment factors obtained using a focus group discussion approach involving manufacturing industry practitioners in Indonesia.

Research opportunities related to the identification of assessment factors for the innovation capability of companies based on intellectual capital are still very open in the future. For example, there are opportunities in the service industry sector, small- and medium-scale industries, and in model or technology-intensive industries. In addition, there are interesting opportunities to conduct research into causal relationships between assessment factors, so that companies have a complete and comprehensive picture of how they can manage innovation capabilities based on intellectual capital.

## 7. Conclusions

This study attempts to determine intellectual capital as an important factor for a firm's innovation capability. This study considered that it can be mobilized in two stages: The identification process and the screening of assessment factors of innovation capabilities, based on intellectual capital. At the identification stage, 18 initial assessment factors were acquired; they were selected based on intellectual capital capabilities that were contextually relevant to the manufacturing industry's current conditions in Indonesia. After passing through the four stages of the screening process using the FDM with a threshold value of $\alpha \geq 0.75$, fourteen factors were accepted and four were rejected. The four factors rejected were human competence, intellectual assets, horizontal linkages, and informal linkages. Of the fourteen assessment factors accepted, four factors have the highest de-fuzzy value, namely: adaptation, innovation behavior, organization culture and climate, and forward linkages. The results of this study showed that three factors for the human capital aspects are important determinants for the assessment of innovation capabilities, based on intellectual capital. There are two factors which have the highest de-fuzzy value; they are adaptation and innovation behavior. The previous research on innovation capability in manufacturing sectors also supported this study; they found that the factors relating to human capital are very influential in developing innovation capabilities.

**Funding:** This research received no external funding.

**Informed Consent Statement:** Informed consent was obtained from all subjects involved in the study.

**Data Availability Statement:** Data is unavailable due to privacy or ethical restriction.

**Conflicts of Interest:** The author declares no conflict of interest.

## Appendix A. A Sample FDM Questionnaire

| Assessment Factors | Description | Linguistic Terms | | | | | | |
|---|---|---|---|---|---|---|---|---|
| | | AU | U | SU | N | SI | I | AI |
| Adaptation | The capability of human resources' adaptation in following the development of the external environment and the speed of response to opportunities measured by the number of options manifested into innovations. | | | | | | | |
| Competence | Company HR capabilities measured by education level, number of certifications, and amount of training. | | | | | | | |
| Innovation behavior | The capability, creativity, and innovation of HR measured by the number of new product ideas and processes sourced from employees. | | | | | | | |
| High motivation and commitment | HR capability in contributing to the company is measured by the level of participation in innovation activities. | | | | | | | |
| IT resources | IT resources' capabilities measured by the availability and reliability of internet and intranet owned by the company. | | | | | | | |
| Technology flexibility | The capability of production technology owned by the company and its future development efforts associated with the company's innovation strategy measured with the novelty and flexibility of production technology. | | | | | | | |
| System and procedure | System capabilities and procedures for supporting innovation activities measured by the availability and effective implementation of SOPs and the number of management system certifications. | | | | | | | |
| Data and information system | Data and information system capabilities measured by the availability and reliability of database systems, data mining systems, and business intelligence. | | | | | | | |
| Culture of innovation | The capability of corporate culture in supporting innovation activities measured by the level of freedom employees must carry out improvements and innovations. | | | | | | | |
| Organization agility | The capability of organizational cohesiveness in responding to and supporting innovation activities measured by the speed of the decision-making process in the organization. | | | | | | | |
| Intellectual assets | Capabilities of the company's intellectual assets measured by the number and application of copyrights, patents, licenses, and trademarks. | | | | | | | |
| Forward linkage | The capability of collaboration with consumers measured by the amount, frequency, and contribution of cooperation. | | | | | | | |
| Backward linkage | The capability of collaboration with suppliers measured by the amount, frequency, and contribution of cooperation. | | | | | | | |
| Horizontal linkage | The capability of collaboration with competitors measured by the amount, frequency, and contribution of cooperation. | | | | | | | |
| Public linkage | The capability of collaboration with universities and the government as measured by the amount, frequency, and contribution of cooperation. | | | | | | | |
| Informal linkage | The capability of collaboration with professional associations is measured by the amount, frequency, and contribution of cooperation. | | | | | | | |
| Cross-functional team | The capability of collaboration between functions within the company measured by the number of work teams across functions and the number of innovation projects between functions. | | | | | | | |
| Sharing and learning | The capability of internal communication and interaction collaboration measured by the number of learning and sharing activities. | | | | | | | |

AU: absolutely unimportant; U: unimportant; SU: slightly unimportant; N: neutral; SI: slightly important; I: important; AI: absolutely important.

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
