# Peer review of "Identifying Key Assessment Factors for a Company’s Innovation Capability Based on Intellectual Capital: An Application of the Fuzzy Delphi Method"

_sustainability, doi:10.3390/su15076001_

Round 1

Reviewer 1 Report

1. Keywords should ideally be phrases of 2-4 words; Research keywords are not carefully selected. It is also necessary to give the reader an adequate explanation after choosing a keyword.

2. What was the lack of available knowledge that led to this research? Research novelty should be clearly stated in the abstract and introduction.

3. It is better to compare the results of this study with other similar studies. Also, the benefits of research are described.

4. In conclusion, the direction of research by other researchers in the future is not specified.

Author Response

Reviewer: 1

Comments and Suggestions for Authors

 No

Reviewer’ Comments

Respond to the comments

All reviewer input has led to corrections as follows:

1

Keywords should ideally be phrases of 2-4 words; Research keywords are not carefully selected. It is also necessary to give the reader an adequate explanation after choosing a keyword.

Four keywords have been selected namely: innovation capability, intellectual capital, sustainable competitiveness, and manufacturing

2

It is better to compare the results of this study with other similar studies. Also, the benefits of research are described.

The abstract and introduction have been changed in accordance with the reviewer's suggestions. Research Novelty has been clearly stated in the abstract and introduction

3

What was the lack of available knowledge that led to this research? Research novelty should be clearly stated in the abstract and introduction

The study results have been compared to other similar studies in the discussion section. The benefits and practical implications of the research have been added to the discussion section.

4

In conclusion, the direction of research by other researchers in the future is not specified

At the end of the conclusion, future research directions have been added.

Reviewer 2 Report

I was able to revise a manuscript entitled “Identifying Key Assesment Factors for a Company's Innovation 2 Capability based on Intellectual Capital: An Application of the 3 Fuzzy Delphi Method”. After the review, I recommend some minor reviews.

Title: Please, correct “assessment”.

Introduction: This is a good section; however, it is hard to associate the text with the current research problem. I recommend a paragraph grounding better the problem and proposing an RQ.

Section 2 name should be a theoretical background and not materials and methods.

Adjust the text in the beginning, there are two innovation words.

Section 2.4 should be section 3.

Results: there are too many values close to 0.75 (e.g., 0.73 and 0.74), it leads me to think if their relevance is below the ones with 0.75. This should be discussed.

Discussion and conclusion: Overall, they are ok. However, the author focuses too much on absolute values, which makes him waste too much effort in an unproductive argumentation. I recommend the author revise such issues and improve such sections.

Author Response

Reviewer: 2

Comments and Suggestions for Authors

  No

Reviewer’ Comments

Respond to the comments

1

Title: Please, correct “assessment”.

Now fixed

2

Introduction: This is a good section; however, it is hard to associate the text with the current research problem. I recommend a paragraph grounding better the problem and proposing an RQ.

The introduction section has been improved by adding one paragraph including the RQ and the advantages of this study compared to previous studies

3

Section 2 name should be a theoretical background and not materials and methods.

This has been changed in accordance with the reviewer's suggestion

4

Adjust the text in the beginning, there are two innovation words.

One of the occurances of the word “innovation” has been deleted

5

Section 2.4 should be section 3.

This has been changed in accordance with the reviewer's suggestion

6

Results: there are too many values close to 0.75 (e.g., 0.73 and 0.74), it leads me to think if their relevance is below the ones with 0.75. This should be discussed.

This input has been explained in the discussion section

7

Discussion and conclusion: Overall, they are ok. However, the author focuses too much on absolute values, which makes him waste too much effort in an unproductive argumentation. I recommend the author revise such issues and improve such sections.

The discussion and conclusion sections have been changed by removing too many absolute values and improving some parts of the discussion and conclusion.

Reviewer 3 Report

Dear Author,

I have read with a great deal of interest your manuscript. Despite the very interesting subject you develop, there are a number of suggestions that you should follow, in order to cover a string of gaps within your research.

Please re-write the Abstract in accordance with the key-sections described by Sustainability | Instructions for Authors (mdpi.com)

Please revise the citation style described within the Journal Instructions for Authors sheets

Literature review out of which the Research question and hypothesis derive are missing

Please revise the text and eliminate typing errors as Innovation Innovation capabilities are..

The Methods section is insufficiently developed; please provide previous literature to support your choice

Please provide details and a short description in regard to how and when was the method applied (please specify an exact date as to when did you collect data – as initial and end points in time), motivate the choice of the SLR, what database did you refer to, justify the choice of the database etc. in regard with the SLR that you used; please provide data in regard to selection criteria/data strings that you used, preliminary and final results.

Similar requirements for FDM. Please provide in the Annexes a sample of the used questionnaire for gathering the expert opinions related to the current method.

Please motivate with literature support the number of experts chosen; please explain the reasoning behind choosing them, with literature support; it is unclear whether the number of experts for each side (practitioners and experts) is sufficient for the current study; please provide the methodology used for the selection of the panel experts members

The results are unclear and insufficiently developed. Please provide literature support for data as for example, why the threshold for α ≥ 0.75.

Please explain why did you choose to adopt the Technology Atlas Project approach; please provide similar choices within the literature; please motivate your choice.

Please provide information in regard with the utility of the FDM in the light of choosing 17 literature factors and only one from the practitioners.

Please provide results in the light of previous findings

Please provide study limitations for the current study

Please provide future developments for the currents study

Please check the English style for the entire manuscript in the light of the intended academical requirements of the Journal 

Author Response

Reviewer: 3

Comments and Suggestions for Authors

No

Reviewer’ Comments

Respond to the comments

1

Please re-write the Abstract in accordance with the key-sections described by Sustainability | Instructions for Authors (mdpi.com)

All reviewer input has led to corrections as follows:

The abstract section has been changed in accordance with the reviewer's suggestion

2

Please revise the citation style described within the Journal Instructions for Authors sheets

The citation style follows the journal instruction for authors

3

Literature review out of which the Research question and hypothesis derive are missing

The literature review section has been adapted to the research question

4

The Methods section is insufficiently developed; please provide previous literature to support your choice

An explanation has added to the methods section in accordance with the reviewer's suggestion

5

Please provide details and a short description in regard to how and when was the method applied (please specify an exact date as to when did you collect data – as initial and end points in time), motivate the choice of the SLR, what database did you refer to, justify the choice of the database etc. in regard with the SLR that you used; please provide data in regard to selection criteria/data strings that you used, preliminary and final results.

Improvements have been made in accordance with the reviewer's input and suggestions

6

Similar requirements for FDM. Please provide in the Annexes a sample of the used questionnaire for gathering the expert opinions related to the current method.

Improvements have been made in accordance with the reviewer's input and suggestions

Round 2

Reviewer 3 Report

-       Dear Author,

I I have read with a great deal of interest your manuscript. Please pay attention to the adjustments necessary as for improving its' quality.

T    The purpose of using the citation style [..] is to avoid specification of Authors within the text.\, unless is absolutely necessary; please provide corrections in this regard;

-        The literature review is insufficiently developed; the  RQ and hypotheses are still missing and are not supported by literature;

-        SLR is a very pretentious method to be used; you need to provide the algorithm, the dates you rallied to, the criteria for selecting the relevant articles,,,,entire content...abstract..keywords etc. (please see the relevant literature in the area and report according to the written methodology). The methodology is inconsistent therefore;

-        Similar observations for FDM;

-        The research does not provide any limitations;

-        Please specify how do you avoid bias within the current research.

-        Additionally from these observations, there are still unanswered questions from the first asessment report that are not worth mentioned again.

Please work slowly but more consistently and improve the current research as o reach the academic level that encourages publishing your manuscript.

Best regards,

Author Response

Reviewer: 3

Comments and Suggestions for Authors

No

Reviewer’ Comments

Responses to the comments

1

The purpose of using the citation style [..] is to avoid specification of Authors within the text.\, unless is absolutely necessary; please provide corrections in this regard

All reviewer input has led to corrections as follows:

Corrections to the citation style have been made following the reviewer's suggestions and in accordance with the journal’s instructions for authors

2

The literature review is insufficiently developed; the  RQ and hypotheses are still missing and are not supported by literature

The literature review section, especially in the section on the factors assessment of innovation capability based on intellectual capital, is now sufficient.

3

SLR is a very pretentious method to be used; you need to provide the algorithm, the dates you rallied to, the criteria for selecting the relevant articles,,,,entire content...abstract..keywords etc. (please see the relevant literature in the area and report according to the written methodology). The methodology is inconsistent therefore

Corrections and additions made according to reviewer suggestions

4

 Similar observations for FDM

Corrections and additions made according to reviewer suggestions

5

 The research does not provide any limitations

Research limitattion has been added at the end of the conclusion

6

Please specify how do you avoid bias within the current research.

We have added an explanation of what was done to avoid bias in this study in the section after the limitations of the research at the end of the conclusion

7

Additionally from these observations, there are still unanswered questions from the first asessment report that are not worth mentioned again

We have answered and responded to all comments from reviewers

Round 3

Reviewer 3 Report

Dear Authors,

Please provide a literature review out of which the research hypotheses and not only question, arise

Please revise the section theoretical background and the bext subtitle; a section title cannot start directly with a subsection title

Within the methodology, the explanation given is not founded and insufficient; you need to lrovide exact database and the exact day you refer to; this type of explanation needs to be specific;

Methodology_ section B you need to actually provide information in regard to the results of the cited literature; a simple citation list is not sufficient;

The tables presented within the results section need to be properly explained within the text; the results need to be discussed in the light of previous literature;

The limitations and future developments from the Conclusions section are not entirely adequate; please verify their relevance in regard to your results;

Please explain how did you avoid the risk of bias for the current methidology;

The manuscript is not clear whether the hypotheses were validated or not; please provide clear data in this regard;

Overall, please pay attention to the methodology section; as it is currently presented, the data appears to have been randomly selected from different search engines, without being clear which ones and when.

Please clarify all these aspects within the manuscript text.

Best regards, 

Author Response

Reviewer: 3 (Round 3)

Comments and Suggestions for Authors

No

Reviewer’ Comments

Response to the comments

1

 Please provide a literature review out of which the research hypotheses and not only question, arise

Please revise the section theoretical background and the bext subtitle; a section title cannot start directly with a subsection title

All the reviewer’s inputs have led to the following corrections:

In the literature review section, major changes have been made by taking into account the reviewer's input.

2

 Within the methodology, the explanation given is not founded and insufficient; you need to lrovide exact database and the exact day you refer to; this type of explanation needs to be specific;

Methodology_ section B you need to actually provide information in regard to the results of the cited literature; a simple citation list is not sufficient;

In the methodology section, we have added more  detail about the stages of the process of selecting papers in accordance with the reviewer's suggestions.

3

 The tables presented within the results section need to be properly explained within the text; the results need to be discussed in the light of previous literature;

In the discussion section, we have now explained the table presented in the results section.

4

The limitations and future developments from the Conclusions section are not entirely adequate; please verify their relevance in regard to your results;

We have added to the limitations and future development sections in accordance with the reviewer's suggestions.

5

 Please explain how did you avoid the risk of bias for the current methidology;

We have improved the methodology to minimize the risk of bias in this paper.

6

 The manuscript is not clear whether the hypotheses were validated or not; please provide clear data in this regard;

Improvements have been made in accordance with the reviewer's input and suggestions.

7

Overall, please pay attention to the methodology section; as it is currently presented, the data appears to have been randomly selected from different search engines, without being clear which ones and when.

The methodology section has had fundamental changes made to it in accordance with the reviewer's suggestions.
